# Energy-Limited Joint Source–Channel Coding of Gaussian Sources over Gaussian Channels with Unknown Noise Level

**DOI:** 10.3390/e25111522

**Published:** 2023-11-06

**Authors:** Omri Lev, Anatoly Khina

**Affiliations:** 1Signals, Information, and Algorithms Laboratory, Massachusetts Institute of Technology (MIT), Cambridge, MA 02139, USA; 2School of Electrical Engineering, Tel Aviv University, Tel Aviv 6997801, Israel; anatolyk@eng.tau.ac.il

**Keywords:** joint source–channel coding, Gaussian channel, infinite bandwidth, energy constraint

## Abstract

We consider the problem of transmitting a Gaussian source with minimum mean square error distortion over an infinite-bandwidth additive white Gaussian noise channel with an unknown noise level and under an input energy constraint. We construct a universal joint source–channel coding scheme with respect to the noise level, that uses modulo-lattice modulation with multiple layers. For each layer, we employ either analog linear modulation or analog pulse-position modulation (PPM). We show that the designed scheme with linear layers requires less energy compared to existing solutions to achieve the same quadratically increasing distortion profile with the noise level; replacing the linear layers with PPM layers offers an additional improvement.

## 1. Introduction

Due to the recent technological advancements in sensing technology and the internet of things, there is a growing demand for low-energy communications solutions. Indeed, since many of the sensors have only limited batteries due to environmental (in the case of energy harvesting) or replenishment limitations, these solutions need to be economical in terms of the utilized energy. Moreover, since each sensor may serve several parties, with each experiencing different conditions, these solutions need to be robust with respect to the noise level.

This problem may be conveniently modeled as the classical setup of conveying *k* independent and identically distributed (i.i.d.) Gaussian source samples with minimum mean square error (MMSE) distortion over a continuous-time additive white Gaussian noise (AWGN) channel under a channel input total energy constraint kE, where *E* is the allowed transmit energy per source sample and unconstrained transmit bandwidth; see Figure 1.

For the encapsulated source–coding problem for a large *k*, the optimal tradeoff between the compression rate *R* and the (per-sample) MMSE distortion *D* [1] (Chapter 13.2) for a memoryless Gaussian source with variance $\sigma_{x}^{2}$ is dictated by the rate–distortion function [1] (Chapter 13.3):(1)$$\begin{array}{c} R\left(D\right)=\left\{\begin{array}{cc} \frac{1}{2}\log\frac{\sigma_{x}^{2}}{D}, & \sigma_{x}^{2}>D,\\ 0, & \sigma_{x}^{2}\leq D \end{array}\right.=\left\{\begin{array}{cc} \frac{1}{2}\log\mathrm{SDR}, & \mathrm{SDR}>1,\\ 0, & \mathrm{SDR}\leq 1, \end{array}\right. \end{array}$$
where $\mathrm{SDR}≜\sigma_{x}^{2}/D$ is the signal-to-distortion ratio (SDR).

For the encapsulated channel-coding problem, since the bandwidth is unconstrained (i.e., grows to infinity), and the allowed energy of the channel input is constrained by *E*, the maximal achievable total reliable rate (in nats) of the entire transmission—the total capacity—is given by [1] (Chapter 9.3)
(2)$$\begin{array}{c} C=\frac{E}{N}=\mathrm{ENR}, \end{array}$$
when the power spectral density of the noise (the *noise level*) *N* is known to the transmitter (and the receiver), and where ENR≜E/N is the energy-to-noise ratio. We note that, in our setting, the transmit energy *E* is fixed regardless of the transmission duration and bandwidth, in contrast to the power-limited setting, in which the energy E=PT grows linearly with the transmission time for a fixed power *P*. To emphasize this, following [2,3] and others, we make use of ENR=E/N to distinguish it from the more common signal-to-noise ratio (SNR), which is defined in the fixed-power scenario as SNR≜P/N.

Returning to the overall problem of conveying *k* i.i.d. source samples of a Gaussian source over a continuous-time AWGN channel subject to an energy constraint (and unconstrained bandwidth), in the limit of a large-source blocklength *k*, the optimal achievable mean square error distortion per source sample is dictated by the celebrated source–channel separation principle [1] (Th. 10.4.1), [4] (Chapter 3.9): R(D)≤C, which upon substituting (Equation 1) and (Equation 2), amounts to
(3)$$\begin{array}{c} D=\sigma_{x}^{2}\cdot e^{-2\mathrm{ENR}}, \end{array}$$
For non-Gaussian continuous memoryless sources, the optimal distortion is bounded as [1] (Prob. 10.8, Th. 10.4.1), [4] (Prob. 3.18, Chapter 3.9)
(4)$$\begin{array}{c} \frac{e^{2h(x)}}{2\pi e}\cdot e^{-2\mathrm{ENR}}\leq D\leq \sigma_{x}^{2}\cdot e^{-2\mathrm{ENR}}, \end{array}$$
where the lower bound stems from Shannon’s lower bound [5], the upper bound holds since a Gaussian source is the “least compressable” source with a given variance under a quadratic distortion measure, and h(x) denotes the differential entropy of a sample of the i.i.d. source *x* [1] (Chapter 8), [4] (Chapter 2.2).

While the optimal performance is known when the transmitter (and the receiver) knows the noise level and k→∞, determining it becomes much more challenging when the noise level is unknown at the transmitter. Indeed, when the transmitter is oblivious of the true noise level, achieving (Equation 3) for all noise levels simultaneously is impossible [6,7]. Instead, one wishes to achieve graceful degradation of the distortion with the noise level, namely, a scheme that would work well for a continuum of all possible noise levels without knowing the true noise level at the transmitter. Since the distortion improves exponentially with the ENR (Equation 3) *when the noise level is known*, in the absence of knowledge of the noise level at the transmitter, one might hope to attain an exponential distortion decay profile with the ENR of the form
(5a)$$\begin{array}{cccccccccc} D & \leq ae^{-b\;\mathrm{ENR}} & & & & & & & \forall \;\mathrm{ENR} & >0 \end{array}$$
for some a,b>0, or, equivalently,
(5b)$$\begin{array}{cccccccccc} D & \leq ae^{-c/N} & & & & & & & \forall N & >0 \end{array}$$
for some a,c>0 and some finite per-sample energy *E*. Köken and Tuncel [7] proved that, unfortunately, this is impossible, namely, no a,b>0 (equivalently a,c>0 and E<∞) exist for which (5a,b) is achievable simultaneously for all ENR>0 (equivalently, for all N>0). Consequently, distortion profiles that deteriorate faster with the noise level need to be sought.

For the case of *finite* bandwidth expansion/compression *B* (and finite power), by superimposing digital successive refinements [8] with a geometric power allocation, Santhi and Vardy [9,10] and Bhattad and Narayanan [11] showed that, in our terms, the distortion improves like ${\mathrm{ENR}}^{-(B-\epsilon )}$ for an arbitrarily small ϵ>0, for *large ENR values*. This suggests that by taking the bandwidth to be large enough, a polynomial decay with an ENR of any finite degree, however large, is achievable, starting from a large enough ENR. In our setting of interest, in which the bandwidth is unconstrained, this means, in turn, that there exists a finite energy *E* for which a *polynomially decaying distortion profile* in *N*
(6a)$$\begin{array}{ccccccccc} D & =\frac{\sigma_{x}^{2}}{1+{\left(\frac{\tilde{E}}{N}\right)}^{L}} & & & & & & & \forall N>0 \end{array}$$
is attainable for any predetermined power 1≤L<∞, however large yet finite, and for any predetermined constant $0<\tilde{E}<\infty$ of our choice, with large enough finite per-sample energy E>0; for the particular choice of $\tilde{E}=a^{1/L}E$ for any constant a>0 of our choice, this is equivalent to
(6b)$$\begin{array}{ccccccccc} D & =\frac{\sigma_{x}^{2}}{1+a\;{\mathrm{ENR}}^{L}} & & & & & & & \forall \;\mathrm{ENR}>0. \end{array}$$

Mittal and Phamdo [12] constructed a different scheme, that works above a certain minimum (not necessarily large) design ENR by sending the digital successive refinements incrementally over non-overlapping frequency bands, and sending the quantization error r of the last digital refinement over the last frequency band.

The scheme of Mittal and Phamdo was subsequently improved by Reznic et al. [6] (see also [13,14], [15] (Chapter 11.1)) by replacing the successive refinement layers with lattice-based Wyner–Ziv coding [16,17], [4] (Chapter 11.3) which, in contrast to the digital layers of the scheme of Mittal and Phamdo, enjoys an improvement in each of the layers with the ENR.

Kokën and Tuncel [7] adopted the scheme of Mittal and Phamdo in the infinite-bandwidth (and infinite-blocklength) setting. Baniasadi and Tuncel [18] (see also [19]) further improved this scheme by allowing the sending of the resulting analog errors of all the digital successive refinements. For the case of a distortion profile that improves quadratically with the ENR (L=2 in (6a,b)), upper and lower bounds were established by Köken and Tuncel [7] and Baniasadi and Tuncel [18] (see also [19]) for the minimum required energy to attain such a profile for all ENR values. For a predetermined value of our choice $\tilde{E}>0$ and a Gaussian source, a quadratic distortion profile ([Disp-formula FD6a-entropy-25-01522]) with a predefined constant $\tilde{E}$ (and L=2) is achievable with a minimal per-sample transmit energy *E* that is bounded as
(7)$$\begin{array}{c} 0.906\tilde{E}\leq E\leq 2.32\tilde{E}. \end{array}$$

A staircase profile was treated by Baniasadi [20] (see also [19]).

However, albeit much progress has been made in determining the minimal required energy to attain polynomially decaying distortion profiles ([Disp-formula FD6a-entropy-25-01522]), with particular emphasis put on the quadratically decaying distortion profile corresponding to the choice L=2, the upper and lower bounds in (Equation 7) remain far apart. Moreover, no (low-delay) schemes for a single-source sample k=1 with graceful degradation of the distortion with the ENR have been proposed.

In this work, we adapt the modulo-lattice modulation (MLM) scheme of Reznic et al. [6] with multiple layers to the infinite-bandwidth setting, and interpret previously decoded layers that are designed for lower ENRs as side information that is known to the receiver but not to the transmitter, which allows, in turn, to apply Wyner–Ziv coding techniques [13], [15] (Chapter 11). By utilizing linear modulation for all the layers, we show that this scheme improves the upper (achievability) bound in (Equation 7). We then replace the analog modulation in (some of) the layers with analog pulse-position modulation (PPM) which was shown to work well for known ENR in [21]. We show that this scheme requires less energy to attain the same quadratic distortion profile compared to the linear-layer-only MLM scheme. Finally, we demonstrate numerically that a low-delay variant of the scheme, which encodes a single-source sample k=1 and uses simple one-dimensional lattices, attains good universal performance with respect to the noise level.

We note that our analytic results rely on the well-established existence of good multi-dimensional lattice codes [15] (to be precisely defined in Section 3), which are used as a building block, along with their known theoretical guarantees. Therefore, our proposed schemes should be understood with this point in mind. That said, for a suboptimal lattice with poorer analytical guarantees, one can similarly calculate the (suboptimal) achievable performance of the scheme. Since lattices work well even in one dimension, we demonstrate the strength of the proposed technique explicitly for this practical scenario using a simple one-dimensional lattice, which amounts to a uniform grid.

The rest of the paper is organized as follows. We introduce the notation that is used in this work in Section 1.1, and formulate the problem setup in Section 2. We provide the necessary background of MLM and analog PPM—the two major building blocks that are used in this work—in Section 3 and Section 4, respectively. We then construct universal schemes with respect to the noise level in Section 5; simulation results of our analysis for good multi-dimensional lattices and of the empirical performance of single-dimensional lattices are provided in Section 6. Finally, we conclude the paper with Section 7 and Section 8 by discussing future research directions and possible improvements.

### 1.1. Notation

N, Z, R, and $R_{+}$ denote the sets of the natural, integer, real, and non-negative real numbers, respectively. With some abuse of notation, we denote tuples (column vectors) by $a^{k}≜{\left(a_{1},\ldots ,a_{k}\right)}^{†}$ for k∈N, and their Euclidean norms by $∥a^{k}∥≜\sqrt{\sum_{i=1}^{k}a_{i}^{2}}$, where ${\left(\cdot \right)}^{†}$ denotes the transpose operation; distinguishing the former notation from the power operation applied to a scalar value will be clear from the context. The i-th element of the vector $a^{k}$ is denoted by $a_{i}$ or by $a\left[i\right]$, where we will use both terms throughout the paper. All logarithms are to the natural base, and all rates are measured in nats. The differential entropy of a continuous random variable with probability density function *f* is defined by $h\left(x\right)≜-\int_{-\infty }^{\infty }f\left(x\right)\log f\left(x\right)dx$ and is measured in nats. The expectation of a random variable (RV) *x* is denoted by $E\left[x\right]$. We denote by ${\left[a\right]}_{L}$ the modulo-*L* operation for a,L∈N, and by ${\left[\cdot \right]}_{\Lambda }$ the modulo-Λ operation [15] (Chapter 2.3) for a lattice Λ [15] (Chapter 2). ⌊·⌋ denotes the floor operation. We denote by $I_{k}$ the *k*-dimensional identity matrix. We denote sets of vectors by capital italic letters, where $A_{b,c}$ stands for a set of *c* vectors, each of length *b*. All the logarithms in this work are to the natural base and all the rates are measured in nats.

## 2. Problem Statement

In this section, we formulate the JSCC setting that will be treated in this work, depicted in Figure 1.

*Source.* The source sequence to be conveyed, $x^{k}\in R^{k}$, comprises *k* i.i.d. samples of a standard Gaussian source, namely, it has mean zero and variance $\sigma_{x}^{2}=1$.

*Transmitter.* Maps the source sequence $x^{k}≜{\left(x_{1},x_{2},\ldots ,x_{k}\right)}^{†}$ to a continuous input waveform $\left\{s_{x^{k}}\left(t\right)|\left|t\right|\leq kT/2\right\}$ that is subject to an energy constraint. (The introduction of negative time instants yields a non-causal scheme. This scheme can be made causal by introducing a delay of size kT/2. We use a symmetric transmission time around zero for convenience):(8)$$\begin{array}{ccccccccc} \int_{-\frac{kT}{2}}^{\frac{kT}{2}}{\left|s(t)\right|}^{2}dt & \leq kE & & & & & & & \forall x^{k}\in R^{k}, \end{array}$$
where *E* denotes the per-symbol transmit-energy. E=PT where *P* is the transmit-power and *T* is the transmission duration.

*Channel.*$s_{x^{k}}$ is transmitted over a continuous-time additive white Gaussian noise (AWGN) channel:(9)$$\begin{array}{ccccccccc} r(t) & =s(t)+n(t), & & & & & & & t\in \left[-\frac{kT}{2},\frac{kT}{2}\right], \end{array}$$
where *n* is a continuous-time AWGN with two-sided spectral density N/2, and *r* is the channel output signal; *N* is referred to as the noise level.

*Receiver.* Receives the channel output signal *r*, and constructs an estimate ${\hat{x}}^{k}$ of $x^{k}$.

*Distortion.* The average quadratic distortion between $x^{k}$ and ${\hat{x}}^{k}$ is defined as
(10)$$\begin{array}{c} D≜\frac{1}{k}E\left[{∥x^{k}-{\hat{x}}^{k}∥}^{2}\right], \end{array}$$
where $∥\cdot ∥$ denotes the Euclidean norm, and the corresponding signal-to-distortion ratio (SDR) by
(11)$$\begin{array}{c} \mathrm{SDR}≜\frac{\sigma_{x}^{2}}{D}=\frac{1}{D}, \end{array}$$
since we assumed $\sigma_{x}^{2}=1$. For non-i.i.d. samples, the variance $\sigma_{x}^{2}$ should be replaced by the effective variance
(12)$$\begin{array}{c} \sigma_{x}^{2}≜\frac{1}{k}E\left[{∥x^{k}∥}^{2}\right], \end{array}$$
which clearly reduces to the (regular) variance in the case of i.i.d. zero-mean samples.

*Regime.* We concentrate on the energy-limited regime, viz. the channel input is not subject to a bandwidth constraint, but rather to an energy constraint *E* per *source symbol* (Equation 8). As explained in the Introduction, the per-source-symbol capacity of the channel (Equation 9) is equal to [1] (Chapter 9.3)
(13)$$\begin{array}{c} C=\mathrm{ENR}, \end{array}$$
where ENR≜E/N, and the capacity is measured in nats; note that the available bandwidth is unconstrained (i.e., infinite).

Since the available bandwidth is unlimited, the receiver can learn the white noise level within any accuracy. Hence, we may assume that the receiver has exact knowledge of the channel conditions. The transmitter is oblivious of the noise level, and needs to accommodate for a continuum of noise levels. Specifically, we will require the distortion to satisfy (6a,b). Throughout most of this work we will concentrate on the setting of infinite blocklength (k→∞). We will also conduct a simulation study for the scalar-source setting (k=1) in Section 6.

## 3. Background: Modulo-Lattice Modulation

The overall scheme, to be introduced and analyzed in Section 5, comprises two major components (in addition to components in the form of interleaving and “Gaussiaization” that are needed for analysis purposes) as depicted in Figure 3:A component that assumes an additive noise vector channel of the same dimension as the source input *k* with unknown noise level, and constructs a layered hybrid digital–analog universal solution with respect to this noise level, where each layer accommodates a different noise level, where an estimator constructed from all the layers that were designed for larger noise levels acts as SI that is known at the receiver;A component that modulates a single analog-source sample over a continuous-time AWGN channel which transforms the channel effectively into a one-dimensional additive channel (one channel use of a discrete-time channel), is designed for a certain noise level but attains graceful improvement if the noise level happens to be better.
Therefore, in this section, we provide the necessary background about the first component. This is a succinct background about lattices and modulo-lattice modulation which is needed to understand the machinery that is used in the proposed solutions in Section 5, along with known performance guarantees that are relevant to this work and are needed for the analysis of the performance guarantees that are claimed in this work. Readers who are less familiar with lattices, lattice coding, and MLM are referred to the well-regarded book of Zamir on this subject [15]. Background about the second component is provided in Section 4.

A *k*-dimensional lattice is a discrete regular array in the Euclidean space $R^{k}$ that is closed under reflection and real addition.

**Definition** **1**(Lattice [15] (Def. 2.1.1))**.**
*A non-degenerate k-dimensional lattice *Λ* is defined by a set of k linearly independent basis (column) vectors $g_{1}^{k},g_{2}^{k},\ldots ,g_{n}^{k}\in R^{k}$, and define the k×k generator matrix G:*
(14)$$\begin{array}{c} G=\left[\begin{array}{cccc} g_{1}^{k} & g_{2}^{k} & \cdots & g_{k}^{k} \end{array}\right]. \end{array}$$
*The lattice *Λ* is composed of all integral combinations of the basis vectors:*
(15a)$$\begin{array}{cc} \Lambda & ≜\left\{\sum_{j=1}^{k}i_{j}g_{j}^{k}|i_{j}\in Z\right\} \end{array}$$
(15b)$$\begin{array}{cc} & =\left\{Gi^{k}|i^{k}\in Z^{k}\right\}. \end{array}$$
*In particular, the origin belongs to the lattice: 0∈Λ.*

Figure 2 provides examples of one- and two-dimensional lattices. A lattice induces a quantization and a partition of the space into cells, with each cell comprising all points that are closest to a specific lattice (quantization) point. These cells are referred to as *Voronoi cells*.

**Definition** **2**(Nearest-neighbor quantizer and Voronoi cell [15] (Chapter 2.2))**.**
*The nearest-neighbor quantizer $Q_{\Lambda }\left(\cdot \right)$ induced by a k-dimensional lattice *Λ* is defined as*
(16)$$\begin{array}{ccccccccc} Q_{\Lambda }\left(x^{k}\right) & ≜\underset{\lambda \in \Lambda }{\mathrm{argmin}}∥x^{k}-\lambda ∥ & & & & & & & \forall x^{k}\in R^{k}. \end{array}$$
*The Voronoi cell $V_{\lambda }$ is the set of all points that are quantized to λ∈Λ:*
(17)$$\begin{array}{c} V_{\lambda }≜\left\{x^{k}\in R^{k}|Q_{\Lambda }\left(x^{k}\right)=\lambda \right\}. \end{array}$$
*$V_{0}$ is referred to as the fundamental Voronoi cell. The breaking of ties in (Equation 16) is carried out in a systematic manner so that the induced Voronoi cells $\left\{V_{\lambda }|\lambda \in \Lambda \right\}$ are congruent. In particular,*
*$V_{\lambda }=V_{0}+\lambda \equiv \left\{x^{k}+\lambda |x^{k}\in V_{0}\right\}$ for λ∈Λ, where the first sum is the Minkowski sum of $V_{0}$ and the singleton $\left\{\lambda \right\}$;**$V_{\lambda_{1}}\cap V_{\lambda_{2}}=\emptyset$ for $\lambda_{1}\neq \lambda_{2}$, where $\lambda_{1},\lambda_{2}\in \Lambda$;**${⋃}_{\lambda \in \Lambda }V_{\lambda }=R^{k}$.*

We next define the modulo-lattice operation with respect to the fundamental Voronoi cell.

**Definition** **3**(Modulo-lattice [15] (Chapter 2.3))**.**
*For a k-dimensional lattice *Λ* with a fundamental Voronoi cell $V_{0}$, the modulo-lattice operation (with respect to $V_{0}$), applied to $x^{k}\in R^{k}$, is defined as*
(18)$$\begin{array}{cc} {\left[x^{k}\right]}_{\Lambda } & ≜x^{k}-Q_{\Lambda }\left(x^{k}\right), \end{array}$$
*namely, the outcome equals the (unique) point that satisfies $x^{k}+\lambda \in V_{0}$ for some λ∈Λ.*

We now define the volume, the second moment, and the normalized second moment of a lattice.

**Definition** **4**(Volume and second moment [15] (Chapter 2 and 3))**.**
*The volume V(Λ) of a k-dimensional lattice *Λ* with fundamental Voronoi cell $V_{0}$ is defined as the volume of $V_{0}$:*
(19)$$\begin{array}{c} V\left(\Lambda \right)≜\mathrm{Vol}\left(V_{0}\right)=\int_{V_{0}}d\left(x^{k}\right). \end{array}$$
*The second moment $\sigma^{2}\left(\Lambda \right)$ of *Λ* is defined as the second moment per dimension of a random variable $d^{k}$ that is uniformly distributed over $V_{0}$:*
(20)$$\begin{array}{cc} \sigma^{2}\left(\Lambda \right) & ≜\frac{1}{k}E\left[{∥d^{k}∥}^{2}\right]=\frac{1}{k}\int_{V_{0}}\frac{{∥x^{k}∥}^{2}}{V(\Lambda )}d\left(x^{k}\right). \end{array}$$
*The normalized second moment G(Λ) of *Λ* is defined as*
(21)$$\begin{array}{c} G\left(\Lambda \right)≜\frac{\sigma^{2}\left(\Lambda \right)}{V^{2/k}\left(\Lambda \right)}. \end{array}$$
*To attain a good MMSE using lattice quantization, G(Λ) should be as close as possible to the normalized second moment of a k-dimensional ball which, in the limit of k→∞, converges to $\frac{1}{2\pi e}$.*

Since the effective source in intermediate layers (this will become clear in the following section) that we would like to transmit and the effective channel noise which is induced by the analog modulations over the continuous-time channel are not Gaussian in general (even after “Gaussianization” which would make them only approximately so), we would need to consider more general source and channel noise vectors that satisfy the following definition of semi-norm ergodicity (SNE).

**Definition** **5**(SNE [22] (Def. 2))**.**
*A sequence in k of random vectors $z^{k}\in R^{k}$ of length k with a limit norm $\sigma_{z}<\infty$. (The original definition of [22] (Def. 2) requires $\sigma_{z}^{\left(k\right)}=\sigma_{z}$ for all k∈N. We use here a more relaxed definition which will prove more convenient in the following section):*
(22)$$\begin{array}{ccccccccc} \sigma_{z}^{\left(k\right)} & ≜\sqrt{\frac{1}{k}E\left[{∥z^{k}∥}^{2}\right]}, & & & & & & & \lim_{k\to \infty }\sigma_{z}^{\left(k\right)}=\sigma_{z}, \end{array}$$
*is SNE if for any ϵ,δ>0, however small, there exists a large enough $k_{0}\in Z$, such that for all $k>k_{0}$*
(23)$$\begin{array}{c} \mathrm{Pr}\left(\frac{1}{k}E\left[{∥z^{k}∥}^{2}\right]>\left(1+\delta \right)\sigma_{z}^{2}\right)\leq \epsilon . \end{array}$$

We are now ready to describe the *k*-dimensional JSCC setting and the MLM technique with side information (SI) for this setting. In the overall solution of Section 5, the analog modulations over the continuous-time channel that will be described in Section 4 will translate the channel into an effective *k*-dimensional additive SNE noise channel (compare also the subfigures of Figure 4 in Section 5. Over this effective channel, MLM with SI will be employed, where we will treat previous source estimators as effective side information (SI) known to the receiver but not to the transmitter [13] and [15] (Chapter 11).

*Source.* Consider a source sequence (equivalently, vector) $x^{k}$ of length *k*,
(24)$$\begin{array}{c} x^{k}=q^{k}+j^{k}, \end{array}$$
where $j^{k}$ is an SI sequence which is known to the receiver but not to the transmitter, and $q^{k}$ is the “unknown part” (at the receiver) with per-element variance
(25)$$\begin{array}{c} \sigma_{q}^{2}≜\frac{1}{k}E\left[{∥q^{k}∥}^{2}\right] \end{array}$$
and is SNE (as a sequence in *k*).

*Transmitter.* Maps $x^{k}$ to a channel input, $m^{k}$, that is subject to a power constraint
(26)$$\begin{array}{c} \frac{1}{k}E\left[{∥m^{k}∥}^{2}\right]\leq P. \end{array}$$

*Channel.* The channel is an additive noise channel:(27)$$\begin{array}{c} y^{k}=m^{k}+z^{k} \end{array}$$
where $z^{k}$ is an SNE noise vector that is uncorrelated with $x^{k}$ and has effective variance
(28)$$\begin{array}{c} \sigma_{z}^{2}≜\frac{1}{k}E\left[{∥z^{k}∥}^{2}\right]. \end{array}$$

The SNR is defined as $\mathrm{SNR}≜P/\sigma_{z}^{2}$; we use here the more common SNR notion in lieu of the ENR notion to emphasize that the channel and the source vectors (equivalently, sequences) in this section are of the same dimension *k*, in contrast to the continuous-time channel of Section 2.

*Receiver.* Receives $y^{k}$, in addition to the SI $j^{k}$, and generates an estimate ${\hat{x}}^{k}\left(y^{k},j^{k}\right)$ of the source $x^{k}$.

The following MLM-based scheme will be employed in the following.**Scheme** **1.** [MLM-based JSCC with SI [13], [15] (Chapter 11)] 

***Transmitter:*** Transmits the signal
(29)$$\begin{array}{c} m^{k}={\left[\eta x^{k}+d^{k}\right]}_{\Lambda } \end{array}$$
where Λ is a lattice with a fundamental Voronoi cell $V_{0}$ and a second moment *P*, η is a scalar scale factor, and $d^{k}$ is a dither vector which is uniformly distributed over $V_{0}$ and is independent of the source vector $x^{k}$; consequently, $m^{k}$ is independent of $x^{k}$ by the so-called crypto lemma [15] (Chapter 4.1).


*
**Receiver:**
*
Receives the signal $y^{k}$ (Equation 27) and generates the signal
(30)$$\begin{array}{c} \begin{array}{c} {\tilde{y}}^{k}={\left[\alpha_{c}y^{k}-\eta j^{k}-d^{k}\right]}_{\Lambda }≜{\left[\eta q^{k}+z_{\mathrm{eff}}^{k}\right]}_{\Lambda } \end{array} \end{array}$$
where $z_{\mathrm{eff}}^{k}≜-\left(1-\alpha_{c}\right)m^{k}+\alpha_{c}z^{k}$ is the equivalent channel noise, and $\alpha_{c}$ is a channel scale factor.Generates an estimate ${\hat{x}}^{k}$:
(31)$$\begin{array}{c} {\hat{x}}^{k}=\frac{\alpha_{s}}{\eta }{\tilde{y}}^{k}+j^{k}, \end{array}$$
where $\alpha_{s}$ is a source scale factor.


When $\eta q^{k}+z_{\mathrm{eff}}^{k}$ in (Equation 30) falls within $V_{0}$, the modulo operation does not come into play, resulting in an effective additive noise channel from $q^{k}$ to ${\tilde{y}}^{k}$. Thus, we want the probability of this “correct lattice decoding” event to be bounded from below by $1-P_{e}$ for some small $P_{e}$. On the other hand, conditioned on the correct lattice decoding event, we want the quantization noise, which is governed by $z_{\mathrm{eff}}^{k}$, and consequently by the shape of $V_{0}$, to have a small normalized second moment to be good for MMSE estimation. The following theorem provides guarantees for the achievable distortion using this scheme and is aggregated from [13], [15] (Chs. 11.3, 6.4, 9.3), and [22] (see also the exposition about correlation-unbiased estimators (CUBEs) in [23]).

**Theorem** **1.**
*The distortion (Equation 10) of Scheme 1 is bounded from above by*

(32)
$$\begin{array}{cc} D & \leq L\left(\Lambda ,P_{e},\alpha_{c}\right)\cdot \tilde{D}+P_{e}\cdot D^{\mathrm{err}}, \end{array}$$

*for $\alpha_{c}\in \left(0,1\right],\alpha_{s}\in \left(0,1\right]$, and η>0 that satisfy*

(33)
$$\begin{array}{c} \frac{\eta^{2}\sigma_{q}^{2}}{P}+\frac{\alpha_{c}^{2}}{\mathrm{SNR}}+{\left(1-\alpha_{c}\right)}^{2}\leq 1, \end{array}$$

*where*

(34)
$$\begin{array}{c} \tilde{D}≜{\left(1-\alpha_{s}\right)}^{2}\sigma_{q}^{2}+\alpha_{s}^{2}\left(\frac{\alpha_{c}^{2}}{\mathrm{SNR}}+{\left(1-\alpha_{c}\right)}^{2}\right)\frac{P}{\eta^{2}}, \end{array}$$

*$D^{\mathrm{err}}$ is the distortion given a lattice-decoding-error event [13] (Equation (Equation 24)) and is bounded from above by*

(35)
$$\begin{array}{c} D^{\mathrm{err}}\leq 4\sigma_{q}^{2}\left(1+\frac{\tilde{L}\left(\Lambda \right)}{\tilde{\alpha }}\right), \end{array}$$

*and the lattice parameters $L\left(\cdot ,\cdot ,\cdot \right)$ and $\tilde{L}\left(\cdot \right)$ are defined as*

(36)
$$\begin{array}{cc} L\left(\Lambda ,P_{e},\alpha_{c}\right) & ≜\min\left\{\ell :\mathrm{Pr}\left(\frac{z_{\mathrm{eff}}^{k}}{\sqrt{\ell }}\notin V_{0}\right)\leq P_{e}\right\}>1, \end{array}$$


(37)
$$\begin{array}{cc} \tilde{L}\left(\Lambda \right) & ≜\frac{\max_{a^{k}\in V_{0}}{∥a^{k}∥}^{2}}{kP}>1. \end{array}$$

*Moreover, for any $P_{e}>0$, however small, and any $\alpha_{c}\in \left(0,1\right]$, there exists a sequence of lattices, $\left\{\Lambda_{k}|k\in N\right\}$, that are good for both channel coding [22] (Def. 4) and mean squared error (MSE) quantization [22] (Def. 5), viz.,*

(38)
$$\begin{array}{c} \begin{array}{cc} \lim_{k\to \infty }L\left(\Lambda_{k},P_{e},\alpha_{c}\right) & =1\\ \lim_{k\to \infty }\tilde{L}\left(\Lambda_{k}\right) & =1, \end{array} \end{array}$$

*respectively, and, therefore, this sequence of lattices achieves a distortion that approaches $\tilde{D}$.*


**Remark** **1.**
*By our definition of SNE sequences, for each finite k the actual variance of the unknown part $\sigma_{q}^{\left(k\right)}$ and the noise variance $\sigma_{z}^{\left(k\right)}$ may be higher than for every k<∞ higher than their asymptotic quantities. Consequently, also the second moment of $\Lambda_{k}$ for every k<∞ would be taken to be higher than its asymptotic value.*

*That said, as k grows to infinity, these slacks become negligible and the performance converges to that of (Equation 32), (Equation 38).*


The following choice of parameters is optimal in the limit of infinite blocklength, k→∞, in the Gaussian case ($q^{k}$ comprises i.i.d. Gaussian samples, $z^{k}$ comprises i.i.d. Gaussian samples) [4] (Chapter 11.3) when the SNR is known.

**Corollary** **1**(Optimal parameters [13], [15] (Chapter 11.3))**.**
*The choice $\alpha_{c}=\alpha_{c}\left(\mathrm{SNR}\right)$, $L=L(\Lambda ,P_{e},\alpha_{c})$, $\tilde{\alpha }=\tilde{\alpha }\left(\alpha_{c},L\right)$, $\alpha_{s}\left(\mathrm{SNR},\tilde{\alpha },\alpha_{c}\right)$, $\eta =\eta (\tilde{\alpha },\sigma_{q}^{2})$ yields a distortion D that is bounded from above as in (Equation 32) with*
(39)$$\begin{array}{cc} \tilde{D} & =\frac{\sigma_{q}^{2}}{1+\tilde{\alpha }\cdot \left(1+\mathrm{SNR}\right)}, \end{array}$$
*where*
(40)$$\begin{array}{cc} \alpha_{c}\left(\mathrm{SNR}\right) & ≜\frac{\mathrm{SNR}}{1+\mathrm{SNR}}, \end{array}$$
(41)$$\begin{array}{cc} \tilde{\alpha }\left(\alpha_{c},L\right) & ≜\max\left(\alpha_{c}-\frac{L-1}{L},0\right)\;, \end{array}$$
(42)$$\begin{array}{cc} \eta (\tilde{\alpha },\sigma_{q}^{2}) & ≜\sqrt{\tilde{\alpha }\frac{P}{\sigma_{q}^{2}}}, \end{array}$$
(43)$$\begin{array}{cc} \alpha_{s}\left(\mathrm{SNR},\tilde{\alpha },\alpha_{c}\right) & ≜\frac{\mathrm{SNR}\cdot \tilde{\alpha }}{\mathrm{SNR}\cdot \tilde{\alpha }+\alpha_{c}}. \end{array}$$
*Moreover, for any $P_{e}>0$, however small, there exists a sequence of lattices $\left\{\Lambda_{k}|k\in N\right\}$ that attains (Equation 38) and, therefore, in the limit k→∞, $\tilde{\alpha }$ and $\alpha_{s}$ above converge to $\alpha_{c}$ and the distortion D approaches $\tilde{D}$, which converges, in turn, to*
(44)$$\begin{array}{c} \tilde{D}=\frac{\sigma_{q}^{2}}{1+\mathrm{SNR}}. \end{array}$$

Consider now the setting of an SNR that is unknown at the transmitter but is known at the receiver. In this case, although the receiver knows the SNR and can, therefore, optimize $\alpha_{c}$ and $\alpha_{s}$ accordingly, the transmitter, being oblivious of the SNR, cannot optimize η for the true value of the SNR. Instead, by setting η in accordance with Corollary 1 for a preset minimal-allowable-design SNR, ${\mathrm{SNR}}_{0}$, Scheme 1 achieves (Equation 44) for $\mathrm{SNR}={\mathrm{SNR}}_{0}$ and improves, albeit sublinearly, with the SNR for $\mathrm{SNR}\geq {\mathrm{SNR}}_{0}$. This is detailed in the next corollary.

**Corollary** **2**(SNR universality)**.**
*Assume that $\mathrm{SNR}\geq {\mathrm{SNR}}_{0}$ for some predefined ${\mathrm{SNR}}_{0}>0$. Then, the choice $L(\Lambda ,P_{e},\alpha_{c}\left({\mathrm{SNR}}_{0}\right))$, $\tilde{\alpha }=\tilde{\alpha }\left(\alpha_{c}\left({\mathrm{SNR}}_{0}\right),L\right)$, and $\eta =\eta (\tilde{\alpha },\sigma_{q}^{2})$ with respect to ${\mathrm{SNR}}_{0}$ (as it cannot depend on the true SNR), and $\alpha_{c}=\alpha_{c}\left(\mathrm{SNR}\right)$ and $\alpha_{s}=\alpha_{s}\left(\mathrm{SNR},\tilde{\alpha },\alpha_{c}\right)$ (may depend on the true SNR) yields a distortion D that is bounded from above, as in (Equation 32) for $\tilde{D}$ that is given in (Equation 39) with $\tilde{\alpha }=\tilde{\alpha }\left(\alpha_{c}\left({\mathrm{SNR}}_{0}\right),L\right)$. Moreover, for any $P_{e}>0$, however small, there exists a sequence of lattices $\left\{\Lambda_{k}|k\in N\right\}$ that satisfies (Equation 38); therefore, in the limit k→∞, $\tilde{\alpha }$ converges to $\alpha_{c}\left({\mathrm{SNR}}_{0}\right)$, $\alpha_{s}$ converges to $\frac{{\mathrm{SNR}}_{0}\left(1+\mathrm{SNR}\right)}{{\mathrm{SNR}}_{0}\left(1+\mathrm{SNR}\right)+1+{\mathrm{SNR}}_{0}}$, and the distortion D approaches $\tilde{D}$, which converges, in turn, to*
(45)$$\begin{array}{cc} \tilde{D} & =\frac{\sigma_{q}^{2}}{1+\mathrm{SNR}}\frac{1}{\frac{1}{1+\mathrm{SNR}}+\frac{{\mathrm{SNR}}_{0}}{1+{\mathrm{SNR}}_{0}}}. \end{array}$$

**Corollary** **3**(Source power uncertainty)**.**
*Assume now additionally that the transmitter is oblivious of the exact power of $q^{k}$, $\sigma_{q}^{2}$, but knows that it is bounded from above by ${\tilde{\sigma }}_{q}^{2}$: $\sigma_{q}^{2}\leq {\tilde{\sigma }}_{q}^{2}$. Then, the distortion is bounded according to (Equation 32), with*
(46)$$\tilde{D}=\frac{{\tilde{\sigma }}_{q}^{2}}{\frac{{\tilde{\sigma }}_{q}^{2}}{\sigma_{q}^{2}}+\tilde{\alpha }\cdot \left(1+\mathrm{SNR}\right)}$$
*for the parameters*
(47)$$\begin{array}{c} \begin{array}{ccccc} \alpha_{c} & =\frac{\mathrm{SNR}}{1+\mathrm{SNR}},\;\tilde{\alpha } & =\tilde{\alpha }\left(\alpha_{c}\left({\mathrm{SNR}}_{0}\right),L\right),\;\eta & =\eta \left(\tilde{\alpha },{\tilde{\sigma }}_{q}^{2}\right),\;\alpha_{s} & =\frac{\tilde{\alpha }\left(1+\mathrm{SNR}\right)}{\frac{{\tilde{\sigma }}_{q}^{2}}{\sigma_{q}^{2}}+\tilde{\alpha }\left(1+\mathrm{SNR}\right)}, \end{array} \end{array}$$
*Moreover, for any $P_{e}>0$, however small, there exists a sequence of lattices $\left\{\Lambda_{k}|k\in N\right\}$ that attains (Equation 38) and, therefore, in the limit of k→∞, $\tilde{\alpha }$ converges to $\alpha_{c}\left({\mathrm{SNR}}_{0}\right)$, $\alpha_{s}$ converges to $\frac{1+\mathrm{SNR}}{\left(1+\mathrm{SNR}\right)+\frac{{\tilde{\sigma }}_{q}^{2}}{\sigma_{q}^{2}}\frac{1+{\mathrm{SNR}}_{0}}{{\mathrm{SNR}}_{0}}}$, and the distortion D is bounded from above in this limit by $\tilde{D}$:*
(48a)$$\begin{array}{cc} D & \leq \tilde{D}+\epsilon \end{array}$$
(48b)$$\begin{array}{cc} & =\frac{{\tilde{\sigma }}_{q}^{2}}{1+\mathrm{SNR}}\cdot \frac{1}{\frac{{\tilde{\sigma }}_{q}^{2}}{\sigma_{q}^{2}}\cdot \frac{1}{1+\mathrm{SNR}}+\frac{{\mathrm{SNR}}_{0}}{1+{\mathrm{SNR}}_{0}}}+\epsilon \end{array}$$
(48c)$$\begin{array}{c} \leq \min\left\{\frac{\sigma_{q}^{2}}{1+{\mathrm{SNR}}_{0}},\frac{{\tilde{\sigma }}_{q}^{2}}{1+\mathrm{SNR}}\frac{1+{\mathrm{SNR}}_{0}}{{\mathrm{SNR}}_{0}}\right\}+\epsilon , \end{array}$$
*where ϵ decays to zero with $P_{e}$. For $\mathrm{SNR}\geq {\mathrm{SNR}}_{0}\gg 1$, the bound ([Disp-formula FD48c-entropy-25-01522]) approaches $\frac{{\tilde{\sigma }}_{q}^{2}}{1+\mathrm{SNR}}$.*

The following result is a simple consequence of Theorem 1 and avoids exact computation of the optimal parameters.

**Corollary** **4**(Suboptimal parameters)**.**
*Assume the setting of Corollary 3 but with $z^{k}$ not necessarily uncorrelated with $m^{k}$, and denote $\mathrm{SDR}=P/\sigma_{z}^{2}$. Then, the distortion is bounded according to (Equation 32) with*
(49)$$\tilde{D}=\frac{{\tilde{\sigma }}_{q}^{2}}{\mathrm{SDR}}$$
*for the parameters $\tilde{\alpha }=\alpha_{c}=\alpha_{s}=1$, $\eta =\eta (1,{\tilde{\sigma }}_{q}^{2})$.*
*We refer to $P/\sigma_{z}^{2}$ by SDR since now $z^{k}$ may depend on $m^{k}$.*


The following property will prove useful in Section 5 when treating non-Gaussian noise through “Gaussianization”.

**Lemma** **1**([24] (Lemmas 6 and 11))**.**
*Let $\left\{\Lambda_{k}|k\in N\right\}$ be a sequence of lattices that satisfies the results in this section, and let $d^{k}$ be a dither that is uniformly distributed over the fundamental Voronoi cell of $\Lambda_{k}$. Then, the probability density function (p.d.f.) of $d^{k}$ is bounded from above as*
(50)$$\begin{array}{ccccccccc} f_{d^{k}}\left(a^{k}\right) & \leq f_{G^{k}}\left(a^{k}\right)e^{\epsilon_{k}k} & & & & & & & \forall a^{k}\in R^{k}, \end{array}$$
*where $f_{G^{k}}$ is the p.d.f. of a vector with i.i.d. Gaussian entries with zero mean and the same second moment P as $\Lambda_{k}$, and $\epsilon_{k}>0$ decays to zero with k.*

## 4. Background: Analog Modulations in the Known-ENR Regime

Following the exposition at the beginning of Section 3 and Figure 3, we concentrate now on the second major component that is used in this work, that of analog modulations for conveying a scalar zero-mean Gaussian source (k=1) over a channel with infinite bandwidth, where both the receiver and the transmitter know the channel noise level, or equivalently, ENR=E/N. To that end, we will review next the analog linear modulation and the analog PPM and will supplement the known results for the latter with a new robustness result for a source distribution that deviates from Gaussianity in Corollary 6.

Consider first analog linear modulation, in which the source sample *x* is linearly transmitted with energy *E*, (under linear transmission, the energy constraint holds only on average, and the transmitted energy is equal to the square of the specific realization of *x*) using some unit-energy waveform
(51)$$\begin{array}{c} s_{x}\left(t\right)=\sqrt{E}\frac{x}{\sigma_{x}}\varphi \left(t\right). \end{array}$$
Note that linear modulation is the same (“universal”) regardless of the true noise level. Signal space theory [25] (Chapter 8.1), [26] (Chapter 2) suggests that a sufficient statistic of the transmission of (Equation 51) over the channel (Equation 9) is the one-dimensional projection *y* of *r* onto φ:(52)$$\begin{array}{c} \begin{array}{cc} y & =\int_{-\frac{T}{2}}^{\frac{T}{2}}\varphi \left(t\right)r\left(t\right)dt=\sqrt{E}\frac{x}{\sigma_{x}}+\sqrt{\frac{N}{2}}z, \end{array} \end{array}$$
where *z* is a standard Gaussian noise variable. The MMSE estimator of *x* from *y* is linear and its distortion is equal to
(53)$$\begin{array}{cc} D & =\frac{\sigma_{x}^{2}}{1+2\mathrm{ENR}}, \end{array}$$
and improves only linearly with the ENR.

Consider now analog PPM, in which the source sample is modulated by the shift of a given pulse rather than by its amplitude (which is the case for analog linear modulation):(54)$$\begin{array}{c} s_{x}\left(t\right)=\sqrt{E}\phi \left(t-x\Delta \right) \end{array}$$
where ϕ is a predefined pulse with unit energy and Δ is a scaling parameter. In particular, the square pulse (Clearly, the bandwidth of this pulse is infinite. By taking a large enough bandwidth *W*, one may approximate this pulse to an arbitrarily high precision and attain its performance within an arbitrarily small gap) is known to achieve good performance. This pulse is given by
(55)$$\begin{array}{cc} \phi (t) & =\left\{\begin{array}{cc} \sqrt{\frac{\beta }{\Delta }}, & \left|t\right|\leq \frac{\Delta }{2\beta },\\ 0, & \mathrm{otherwise}, \end{array}\right. \end{array}$$
for a parameter β>1, which is sometimes referred to as *effective dimensionality*. Clearly, T=Δ+Δ/β.

The optimal receiver is the MMSE estimator $\hat{x}$ of *x* given the entire output signal: (56)$$\begin{array}{c} {\hat{x}}^{\mathrm{MMSE}}=E\left[x|r\right]. \end{array}$$
The following theorem provides an upper bound on the achievable distortion of this scheme using (suboptimal) maximum a posteriori (MAP) decoding, which is given by
(57)$$\begin{array}{cc} {\hat{x}}^{\mathrm{MAP}} & =\underset{a\in R}{\mathrm{argmax}}\left\{R_{r,\phi }\left(a\Delta \right)-\frac{N}{4\sqrt{E}}a^{2}\right\}, \end{array}$$
where
(58a)$$\begin{array}{c} \begin{array}{cc} R_{r,\phi }\left(\hat{x}\Delta \right) & ≜\int_{-\infty }^{\infty }r\left(t\right)\phi \left(t-\hat{x}\Delta \right)dt=\sqrt{E}R_{\phi }\left((x-\hat{x})\Delta \right)+\sqrt{\frac{\beta }{\Delta }}\int_{\hat{x}\Delta -\frac{\Delta }{2\beta }}^{\hat{x}\Delta +\frac{\Delta }{2\beta }}n\left(t\right)dt, \end{array} \end{array}$$
is the (empirical) cross-correlation function between *r* and ϕ with lag (displacement) $\hat{x}\Delta$, and
(58b)$$\begin{array}{c} \begin{array}{cc} R_{\phi }\left(\tau \right) & =\int_{-\infty }^{\infty }\phi \left(t\right)\phi \left(t-\tau \right)dt=\left\{\begin{array}{cc} 1-\frac{\left|\tau \right|}{\frac{\Delta }{\beta }}, & \left|\tau \right|\leq \frac{\Delta }{\beta }\\ 0, & \mathrm{otherwise} \end{array}\right. \end{array} \end{array}$$
is the autocorrelation function of ϕ with lag τ.

**Remark** **2.**
*Since a Gaussian source has infinite support, the required overall transmission time T is infinite. Of course, this is not possible in practice. Instead, one may limit the transmission time T to a very large—yet finite—value. This will incur a loss compared to the the bound that will be stated next; this loss can be made arbitrarily small by taking T to be large enough.*


**Theorem** **2**([21] (Prop. 2))**.**
*The distortion of the MAP decoder (Equation 57) of a standard Gaussian scalar source transmitted using analog PPM with a rectangular pulse is bounded from above by*
(59)$$\begin{array}{c} D\leq D_{S}+D_{L} \end{array}$$
*with*
$$\begin{array}{cc} \; & D_{L}≜2\beta \sqrt{\mathrm{ENR}}e^{-\frac{\mathrm{ENR}}{2}}(1+3\sqrt{\frac{2\pi }{\mathrm{ENR}}}+\frac{12e^{-1}}{\beta \sqrt{\mathrm{ENR}}}+\frac{8e^{-1}}{\sqrt{8\pi }\beta }\\ & \;+\sqrt{\frac{8}{\pi \mathrm{ENR}}}+\frac{12^{\frac{3}{2}}e^{-\frac{3}{2}}}{\beta \sqrt{32\pi \mathrm{ENR}}})+\beta \sqrt{8\pi }e^{-\mathrm{ENR}}\left(1+\frac{4e^{-1}}{\beta \sqrt{2\pi }}\right),\\ & D_{S}≜\frac{\frac{13}{8}+\sqrt{\frac{2}{\beta }}\left(\sqrt{2\beta \mathrm{ENR}}-1\right)\cdot e^{-{\left(\sqrt{\mathrm{ENR}}-\frac{1}{\sqrt{2\beta }}\right)}^{2}}}{{\left(\sqrt{\beta \mathrm{ENR}}-\frac{1}{\sqrt{2}}\right)}^{4}}+\frac{e^{-\beta \mathrm{ENR}}}{\beta^{2}}, \end{array}$$
*bounding the small- and large-error distortions, assuming βENR>1/2. In particular, in the limit of large ENR, and β that increases monotonically with ENR,*
(60)$$\begin{array}{c} D\leq \left({\tilde{D}}_{S}+{\tilde{D}}_{L}\right)\left\{1+o\left(1\right)\right\} \end{array}$$
*where*
(61)$$\begin{array}{cc} {\tilde{D}}_{S} & ≜\frac{13/8}{{\left(\beta \mathrm{ENR}\right)}^{2}}, \end{array}$$
(62)$$\begin{array}{cc} {\tilde{D}}_{L} & ≜2\beta \sqrt{\mathrm{ENR}}\cdot e^{-\frac{\mathrm{ENR}}{2}}, \end{array}$$
*and o(1)→0 in the limit of ENR→∞.*

**Remark** **3.**
*For a fixed β, the distortion improves quadratically with the ENR. This behavior will prove useful in the next section, where we construct schemes for the unknown-ENR regime.*


Setting $\beta ={\left(\frac{13}{8}\right)}^{\frac{1}{3}}{\left(\mathrm{ENR}\right)}^{-\frac{5}{6}}e^{\frac{\mathrm{ENR}}{6}}$ in (Equation 60) of Theorem 2 yields the following asymptotic performance.

**Corollary** **5**([21] (Th. 2))**.**
*The achievable distortion of a standard Gaussian scalar source transmitted over an energy-limited channel with a known ENR is bounded from above as*
(63)$$\begin{array}{cc} D & \leq 3\cdot {\left(\frac{13}{8}\right)}^{\frac{1}{3}}\;e^{-\frac{\mathrm{ENR}}{3}}\cdot {\left(\mathrm{ENR}\right)}^{-\frac{1}{3}}\cdot \left\{1+o(1)\right\}, \end{array}$$
*where o(1)→0 as ENR→∞.*

The following corollary, whose proof is available in Appendix A, states that the (bound on the) distortion is continuous in the source p.d.f. around a Gaussian p.d.f. Such continuity results of the MMSE estimator in the source p.d.f. are known [27]. Next, we prove the required continuity directly for our case of interest with an additional technical requirement on the deviation from a Gaussian p.d.f.; this result will be used in conjunction with a non-uniform variant of the Berry–Esseen theorem in Section 5.

**Corollary** **6.**
*Consider the setting of Theorem 2 for a source p.d.f. that satisfies*

(64)
$$\begin{array}{cccccccccc} \left|f_{x}\left(a\right)-f_{G}\left(a\right)\right| & \leq \epsilon \delta_{f}\left(a\right), & & & & & & & \forall a & \in R, \end{array}$$

*where ϵ>0, $f_{G}$ is the standard Gaussian p.d.f., and $\delta_{f}$ is a symmetric absolutely continuous non-negative bounded function with unit integral $\int_{\infty }^{\infty }\delta_{f}\left(a\right)da=1$, that is monotonically decreasing for x>0 (and for x<0, by symmetry) and satisfies $\delta_{f}\left(x\right)\in o\left(x^{-4}\right)$; thus, there exists H<∞ such that*

(65)
$$\begin{array}{cccccccc} \delta_{f}\left(x\right)\leq \frac{H}{{\left(1+x\right)}^{4}}, & & & & & & & \forall x\in R. \end{array}$$


*Then, the distortion of the decoder that applies the decoding rule (Equation 57) is bounded from above by*

(66)
$$\begin{array}{c} D\leq D_{G}+\epsilon C, \end{array}$$

*where $D_{G}=D_{S}+P_{L}D_{L}$ denotes the bound on the distortion for a standard Gaussian source of Theorem 2, and C<∞ is a non-negative constant that depends on $\delta_{f}$. (This is no longer the MAP decoding rule since $f_{x}$ is no longer a Gaussian p.d.f.).*


## 5. Main Results

In this section, we construct JSCC solutions for the unknown-ENR communication problem. As already explained at the beginning of Section 3, the proposed solution, which is depicted in Figure 3 (cf. Figure 1), is composed of two major components:Aa layered MLM-based component that works well for a continuum of possible noise levels over *k*-dimensional additive SNE noise channels, where each layer accommodates a different noise level, with layers of lower noise levels acting as SI in the decoding of subsequent layers;An analog modulation component that is designed for a particular ENR of the continuous-time channel but improves for high ENRs and induces a *k*-dimensional additive SNE noise channel for the first component.

Following the exposition in the introduction, since an exponential improvement with the ENR cannot be attained in this setting for an infinite number of noise levels let alone a continuum thereof [7], following [7,18], we consider polynomially decaying profiles (6a,b).

We first show, in Section 5.1, that replacing the successive refinement coding of [7,18] with MLM (Wyner–Ziv coding) with *linear* layers results in better performance in the infinite-bandwidth setting (paralleling the results of the bandwidth-limited setting [6]).

In Section 5.2, we replace the last layer with an analog PPM one, which improves quadratically with the ENR (L=2 in ([Disp-formula FD6b-entropy-25-01522])) above the design ENR (recall Remark 3).

In principle, despite analog PPM attaining a gracious quadratic decay with the ENR (recall Remark 3) only above a predefined design ENR, since the distortion is bounded from above by the (finite) variance of the source, it attains a quadratic decay with the ENR for all $\mathrm{ENR}\in R_{+}$, or equivalently, for all $N\in R_{+}$ and L=2 in (6a,b).

That said, the performance of analog PPM deteriorates rapidly when the ENR is below the design ENR of the scheme, meaning that the minimum energy required to obtain ([Disp-formula FD6a-entropy-25-01522]) with L=2 and a given $\tilde{E}$ is large. To alleviate this, we use the above-mentioned layered MLM scheme. Furthermore, to achieve higher-order improvement with the ENR (L>2 in (6a,b)), multiple layers in the MLM scheme need to be employed.

We now present a simplified variant of the general scheme that is considered throughout this section. This variant is also depicted in Figure 4a. The full scheme, which incorporates interleaving for analytical purposes, is available in Appendix B and depicted in Figure A1. **Scheme** **2** (MLM-based)**.**


***M*-Layer Transmitter:**



*First layer (i=1):*
Transmits each of the entries of the vector $x^{k}$ over the channel (Equation 9) linearly (Equation 51):
$$\begin{array}{ccc} s_{1;\ell }\left(t\right) & ≜s\left(t+\left(\ell -1\right)T\right)=\sqrt{\frac{E_{1}}{T}}\frac{x_{\ell }}{\sigma_{x}}\varphi \left(t\right),\; & t\in [0,T), \end{array}$$
for ℓ∈{1,…,k}, where φ is a continuous unit-norm (i.e., unit-energy) waveform that is zero outside the interval [0,T], say ϕ of (Equation 55), $E_{1}\in \left[0,E\right]$ is the allocated energy for layer 1, and *E* is the total available energy of the scheme.


*Other layers:* For each i∈{2,…,M}:
Calculates the *k*-dimensional tuple
(67)$$\begin{array}{cc} m_{i}^{k} & ={\left[\eta_{i}x^{k}+d_{i}^{k}\right]}_{\Lambda_{i}}\;, \end{array}$$
where $m_{i}^{k}={\left[\begin{array}{cccc} m_{i;1} & m_{i;2} & \ldots & m_{i;k} \end{array}\right]}^{†}$, and $m_{i;\ell }$ denotes the ℓth entry of $m_{i}^{k}$; $\eta_{i}$, $d_{i}^{k}$, and $\Lambda_{i}$ take the roles of the $\eta ,d^{k}$, and Λ of Scheme 1, and are tailored for each layer *i*; $\Lambda_{i}$ is chosen to have unit second moment.For each ℓ∈{1,…,k}, views $m_{i;\ell }$ as a scalar-source sample, and generates a corresponding channel input,
(68)$$\begin{array}{ccccccccc} s_{i;\ell }\left(t\right) & ≜s\left(t+(\ell -1)T+(i-1)kT\right), & & & & & & & t\in [0,T), \end{array}$$
using a scalar JSCC scheme with a predefined energy $E_{i}\geq 0$ that is designed for a predetermined ${\mathrm{ENR}}_{i}$, or equivalently, $N_{i}=E_{i}/{\mathrm{ENR}}_{i}$, such that $\sum_{i=1}^{M}E_{i}=E$ and $N_{2}>N_{3}>\cdots >N_{M}>0$.

***Receiver:*** Receives the channel output signal *r* (Equation 9), and recovers the different layers as follows.

*First layer (i=1):* For each ℓ∈{1,…,k}:

Recovers the MMSE estimate ${\hat{x}}_{1;\ell }$ of $x_{\ell }$ given $\left\{r_{1;\ell }\left(t\right)|t\in \left[0,T\right)\right\}$, where $r_{1;\ell }\left(t\right)≜r\left(t+\left(\ell -1\right)T\right)$.If the true noise level *N* satisfies $N>N_{2}$, sets the final estimate ${\hat{x}}_{\ell }$ of $x_{\ell }$ to ${\hat{x}}_{1;\ell }$ and stops. Otherwise, determines the maximal layer index 𝚥∈{2,…,M} for which $N\leq N_{𝚥}$ and continues to process the other layers.

*Other layers:* For each i∈{2,…,j} in ascending order:
For each ℓ∈{1,…,k}, uses the receiver of the scalar JSCC scheme to generate an estimate ${\hat{\tilde{m}}}_{i;\ell }$ of ${\tilde{m}}_{i;\ell }$ from $\left\{r_{i;\ell }\left(t\right)|t\in \left[0,T\right)\right\}$, where
(69)$$\begin{array}{c} r_{i;\ell }\left(t\right)≜r\left(t+(\ell -1)T+(i-1)kT\right). \end{array}$$Using the effective channel output ${\hat{m}}_{i}^{k}$ (that takes the role of $y^{k}$ in Scheme 1) with SI ${\hat{x}}_{i-1}^{k}$, generates the signal
(70)$$\begin{array}{cc} {\tilde{y}}_{i}^{k} & ={\left[\alpha_{c}^{\left(i\right)}{\hat{m}}_{i}^{k}-\eta_{i}{\hat{x}}_{i-1}^{k}-d_{i}^{k}\right]}_{\Lambda_{i}}, \end{array}$$
as in (Equation 30) of Scheme 1, where $\alpha_{c}^{\left(i\right)}$ is a channel scale factor.Constructs an estimate ${\hat{x}}_{i}^{k}$ of $x^{k}$:
(71)$$\begin{array}{cc} {\hat{x}}_{i}^{k} & =\frac{\alpha_{s}^{\left(i\right)}}{\eta_{i}}{\tilde{y}}_{i}^{k}+{\hat{x}}_{i-1}^{k}, \end{array}$$
as in (Equation 31) of Scheme 1, where $\alpha_{s}^{\left(i\right)}$ is a source scale factor. The final estimate if ${\hat{x}}^{k}={\hat{x}}_{𝚥}^{k}$.

**Remark** **4**(Interleaving)**.**
*To guarantee independence between all the noise entries ℓ∈{1,…,k}, we use interleaving in the full scheme, which is described in Appendix B in (A8) and (A11). We note that this operation is used to simplify the proof that the resulting noise vector is SNE (recall Definition 5).*

**Remark** **5**(Gaussianization)**.**
*To use the analysis of Section 4 of analog PPM for a Gaussian source, we multiply the vectors $m_{i}^{k}$ by orthogonal matrices $H_{i}$ that effectively “Gaussianize” its entries, as shown in the full description of the scheme in Appendix B, in (A8) and (A11). In particular, this is achieved by a Walsh–Hadamard matrix $H_{i}$ by appealing to the central limit theorem; a similar choice was previously proposed by Feder and Ingber [28], and by Hadad and Erez [29], where in the latter, the columns of the Walsh–Hadamard matrix were further multiplied by i.i.d. Rademacher RVs to achieve near-independence between multiple descriptions of the same source vector (see [29,30,31] for other ensembles of orthogonal matrices that achieve a similar result). Interestingly, the multiplication by the orthogonal matrices $H_{i}^{-1}=H_{i}^{†}$ (since Walsh–Hadamard matrices are symmetric, they further satisfy $H_{i}^{†}=H_{i}$) Gaussianizes the effective noise incurred at the outputs of the analog PPM JSCC receivers.*

**Remark** **6**(JSCC-induced channel)**.**
*The continuous-time JSCC transmitter and receiver over the infinite-bandwidth AWGN channel induce an effective additive-noise channel of better effective SNR and source bandwidth. Over this induced channel, the MLM transmitter and receiver are then employed. This interpretation is depicted in Figure 4b, with ${\tilde{n}}_{i}^{k}$ representing the effective additive noise vectors.*

We next provide analytic guarantees for this scheme for linear and analog PPM layers in Section 5.1 and Section 5.2, respectively, in the infinite-blocklength regime. In Section 6, we compare the analytic and empirical performance of these schemes in the infinite-blocklength regime, as well as comparing the empirical performance of these schemes for a single-source sample. The treatment of the infinite-blocklength regime pertains to the full scheme as presented in Appendix B. The comparison for a single-source sample, uses the simplified variant of Scheme 2.

### 5.1. Infinite-Blocklength Setting with Linear Layers

We start with analyzing the performance of the scheme where all the *M* layers are transmitted linearly and *M* is large; we concentrate on the setting of an infinite-source blocklength (k→∞) and derive an achievability bound on the minimum energy that achieves a polynomial distortion profile (6a,b). A constructive proof of the next theorem is available in Appendix C. In particular, this proof specifies all the scheme parameters, such as the energy allocated to each layer and the minimal noise level it is designed for.

**Theorem** **3.**
*Choose a decaying order L>1, a design parameter $\tilde{E}>0$, and a minimal noise level $N_{\min}>0$, however small. Then, a distortion profile ([Disp-formula FD6a-entropy-25-01522]) with L and $\tilde{E}$ is achievable for all noise levels $N>N_{\min}$ for any transmit energy E that satisfies*

(72)
$$\begin{array}{c} E>\delta_{\mathrm{lin}}\left(L\right)\tilde{E}, \end{array}$$

*for a large-enough-source blocklength k, where*

(73)
$$\begin{array}{cc} \delta_{\mathrm{lin}}\left(L\right) & ≜\frac{1}{2}\cdot \min_{\left(\alpha ,x\right)\in R_{+}^{2}}\left\{{\left(\frac{e^{\alpha }}{x}\right)}^{L-1}+\frac{x}{2}\left(e^{\alpha L}-1\right)\left(1+\sqrt{1+\frac{4e^{\alpha \left(L+1\right)}}{{\left(1-e^{\alpha L}\right)}^{2}}}\right)\frac{e^{-2\alpha }}{1-e^{-\alpha }}\right\}. \end{array}$$

*In particular, the choice $\left(x,\alpha \right)=\left(0.898,0.666\right)$ achieves a quadratic decay (L=2) for any transmit energy E that satisfies*

(74)
$$\begin{array}{c} E>2.167\tilde{E}, \end{array}$$

*for a large-enough-source blocklength k.*


We note that already this variant of the scheme offers an improvement compared to the hitherto best-known upper (achievability) bound of (Equation 7).

The choice of the minimal noise level $N_{\min}$ dictates the number of layers *M* that need to be employed: the lower $N_{\min}$ is, the more layers *M* need to be employed.

**Remark** **7.**
*In the proof in Appendix C, we use an exponentially decaying noise level series $N_{i}=\Delta e^{-\alpha \left(i-1\right)}$, which facilitates the analysis. Nevertheless, any other assignment that satisfies the profile requirement and energy constraint is valid and may lead to better performance; for further discussion, see Section 7.*


### 5.2. Infinite-Blocklength Setting with Analog PPM Layers

In this section, we concentrate on the setting of an infinite-source blocklength (k→∞) and a quadratically decaying profile (L=2 in (6a,b)) using analog PPM.

To that end, we use a sequence of M−1 linear JSCC layers as in Section 5.1, with only the last layer replaced by an analog PPM one; since analog PPM improves quadratically with the ENR (recall Remark 3), *M* need not go to infinity to attain a quadratically decaying profile.

**Theorem** **4.**
*Choose a design parameter $\tilde{E}>0$, and a minimal noise level $N_{\min}>0$, however small. Then, a quadratic profile (L=2) ([Disp-formula FD6a-entropy-25-01522]) with $\tilde{E}$ is achievable for all noise levels $N>N_{\min}$ for any transmit energy E that satisfies*

(75)
$$\begin{array}{c} E>1.961\tilde{E}, \end{array}$$

*for a large-enough-source blocklength k.*


This theorem, whose proof is available in Appendix D, offers a further improvement over the upper bounds in (Equation 7) and Theorem 3 for a quadratic profile. Again, the proof of Theorem 4 in Appendix D is constructive and details the scheme parameters, such as the energy allocated to each layer and the minimal noise level, it is designed for.

**Remark** **8.**
*Replacing all layers but the first layer with analog PPM ones should yield better performance, but complicates the analysis. Moreover, a similar analysis to that of Theorem 3 for L≠2 may be devised, but for L>2 would require multiple layers as the distortion of analog PPM decays only quadratically. Both of these analyses are left for future research.*


## 6. Simulations

In Section 6.1, we first compare the analytical results of Theorems 3 and 4 to the prior art in the infinite-blocklength regime (k→∞). We further optimize the parameters in Theorem 4 empirically and show a further improvement, which suggests, in turn, a slack in our analysis. In Section 6.2, we evaluate the performance of Scheme 2 empirically, using a Monte Carlo simulation, for a single-source sample (k=1) of a uniform source, and compare the performance of a scheme with all linear layers to those of a scheme that incorporates an analog PPM layer.

### 6.1. Analytical Performance Comparison in the Infinite-Blocklength Regime

We first consider the infinite-blocklength regime (k→∞) for a Gaussian source and a quadratic profile (L=2 in (6a,b)), for which we have derived analytical guarantees in Section 5.1 and Section 5.2. Figure 5 depicts the accumulated energy of the employed layers at the receiver of Section 5 and the achievable distortion as functions of $\tilde{E}/N$, along with the desired quadratic distortion profile ([Disp-formula FD6a-entropy-25-01522]) (with L=2) for $N_{\min}\to 0$ for linear layers with the energy allocation to the different layers, as per the proof of Theorem 3 in Appendix C; and M−1 linear layers with a final analog PPM layer (Theorem 4) for both the energy allocation for M=7 layers, which is available in the proof of Theorem 4 in Appendix D and relies on the bound on the analog PPM performance, and for M=2 layers, with an empirically evaluated performance of analog PPM allocation.

This figure clearly demonstrates the gain due to introducing an analog PPM layer. Interestingly, the empirically evaluated analog PPM curve shows that only two layers are needed when the second layer is an analog PPM one, meaning that the seven layers needed in the proof of Theorem 4 are an artifact of the slack in our analytic bounds.

To derive the performance of the scheme with linear layers, we evaluated the energy allocation in the proof of Theorem 3 in Appendix C in (16) directly for the optimized energy allocation $E_{i}=\Delta e^{-\alpha i}$ with Δ=0.975 and α=0.65. To derive the analytical performance of Theorem 4, we used the energy allocation from its proof in Appendix D, while for the empirical performance, optimizing over the energy allocation yielded $E_{1}=0.975\tilde{E},E_{2}=0.5904\tilde{E}$. The Matlab code package and the specific script that was used for generating Figure 5, along with all the scheme parameters and analog PPM empirical evaluation, are available in [32].

### 6.2. Empirical Performance Comparison for a Single-Source Sample

We move now to the uniform scalar-source setting (k=1) and a quadratic profile. The analysis of Section 5 in the scalar setting is difficult. We, therefore, evaluate its performance empirically for both variants of the scheme: with linear layers, and with one linear layer and one analog PPM layer (two layers suffice in this setting as well). In Figure 6, we depict again the accumulated energy of the employed layers at the receiver of Section 5 and the achievable distortion as functions of $\tilde{E}/N$ for both variants of the scheme, along with the desired quadratic distortion profile ([Disp-formula FD6a-entropy-25-01522]) (with L=2) for $N_{\min}\to 0$.

For the variant with linear layers only, an energy allocation of $E_{i}/\tilde{E}=\Delta e^{-\alpha i}$ with Δ=0.9 and α=0.64 was used. For the variant with an analog PPM layer, an energy of $E_{1}=0.9\tilde{E}$ was allocated to the (first) linear layer, and an energy of $E_{2}=0.346\tilde{E}$ was allocated to the (second) analog PPM layer. The lattice inflation factor η was chosen as the minimizer of a variant of (Equation 32) under the assumption that the noise is Gaussian, namely,
(76)$$\begin{array}{c} \eta^{*}=\underset{\eta }{\mathrm{argmin}}\;\left\{P_{e}\cdot {\left(2\Delta \right)}^{2}+\left(1-P_{e}\right)\cdot \frac{1}{\eta^{2}N}\right\}, \end{array}$$
where $P_{e}=2Q\left(\frac{\Delta }{\sqrt{\eta^{2}D\left(\mathrm{ENR}\right)+N}}\right)$, *N* is the noise level, Δ is the modulo size that was chosen to be $\sqrt{12}$, and D(ENR) is the average distortion that corresponds to the last transmitted layer.

As in the infinite-blocklength regime, here too utilizing analog PPM provides better performance compared to a linear-only scheme. Again, the Matlab code package and the specific script that was used for generating Figure 6, along with all the scheme parameters and empirical evaluations, are available in [32].

## 7. Summary and Discussion

In this work, we studied the problem of JSCC over an energy-limited channel with unlimited bandwidth and/or transmission time when the noise level is unknown at the transmitter. We showed that MLM-based schemes outperform the existing schemes thanks to the improvement in the performance of all layers (including the preceding layers that act as SI) with the ENR. By replacing (some of the) linear layers with analog PPM ones, further improvement was achieved. We further demonstrated numerically that the MLM-layered scheme works well in the scalar-source regime.

We also note that a substantial gap remains between the lower bound in (Equation 7) and the upper bound of Theorem 4 for the energy required to achieve a quadratic profile ((6a,b) with L=2). In Section 8, several ways to close this gap are described.

We note that, although we assumed that both the bandwidth and the time are unlimited, the scheme and analysis presented in this work carry over to the setting where one of the two is bounded as long as the other one is unlimited, with little adjustment.

## 8. Future Research

Consider first the remaining gap between the lower and upper bounds. As demonstrated in Section 6, the upper (achievability) bound on the performance of analog PPM is not tight and calls for further improvement thereof. This step is currently under intense investigation, along with improvement via companding of the presented analog PPM variant in this work as well as via other choices of energy allocation (see Remark 7). Furthermore, the optimization was performed numerically and for a particular form of noise levels of an exponential form (recall Remark 7). We believe that a systematic optimization procedure could shed light on the weaknesses of our scheme and provide further improvements in the overall performance. On the other hand, the outer bounds of [18] are based on specific choices of sequences of noise levels. Therefore, further improvement might be achieved by other choices and calls for further research.

We have also shown that the MLM scheme performs well in the scalar-source regime; it would be interesting to derive analytical performance guarantees for this regime.

Finally, since MLM utilizes well source SI at the receiver and channel SI at the transmitter [13,14], [15] (Chs. 10–12), the proposed scheme can be extended to limited-energy settings, such as universal transmission with respect to the noise level and the SI quality at the receiver [33] and the dual problem of the one considered in this work of universal transmission with respect to the noise level with near-zero bandwidth [34].

## Figures and Tables

**Figure 1 entropy-25-01522-f001:**
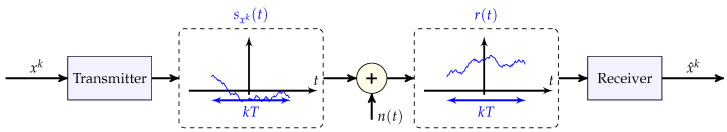
JSCC of *k* samples of a Gaussian source, $x^{k}$, over a continuous-time bandwidth-unlimited AWGN channel subject to an energy constraint (Equation 8). The noise level of the noise process *n* is assumed to be known only to the receiver but not to the transmitter. The transmitter maps the *k* source samples $x^{k}\in R^{k}$ into a continuous-time channel input $\left\{s_{x^{k}}\left(t\right)|\;|t|\leq kT/2\right\}$ (and arbitrarily large bandwidth). The receiver constructs an estimate ${\hat{x}}^{k}$ of $x^{k}$ from the continuous-time channel output $\left\{r(t)|\;|t|\leq kT/2\right\}$.

**Figure 2 entropy-25-01522-f002:**
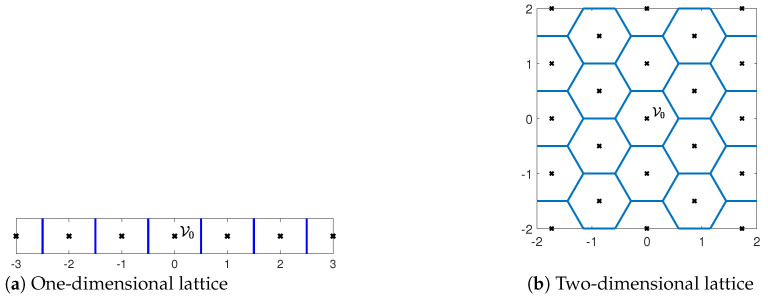
Examples of one- and two-dimensional lattices with generator matrices G=[1] and $G=\left[\begin{array}{cc} \frac{\sqrt{3}}{2} & 0\\ \frac{1}{2} & 1 \end{array}\right]$, respectively. The lattice points are marked by black crosses, whereas the Voronoi cell partitions are marked by blue lines.

**Figure 3 entropy-25-01522-f003:**
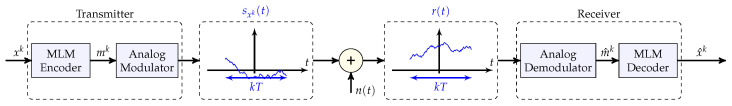
High-level description of the JSCC scheme (Scheme 2). Our construction consists of a concatenation of an MLM encoder and an analog modulation that maps the encoded signals into a continuous-time waveform. On the receiver, we apply the inverse operations by first demodulating the analog signal and then use an MLM decoder.

**Figure 4 entropy-25-01522-f004:**
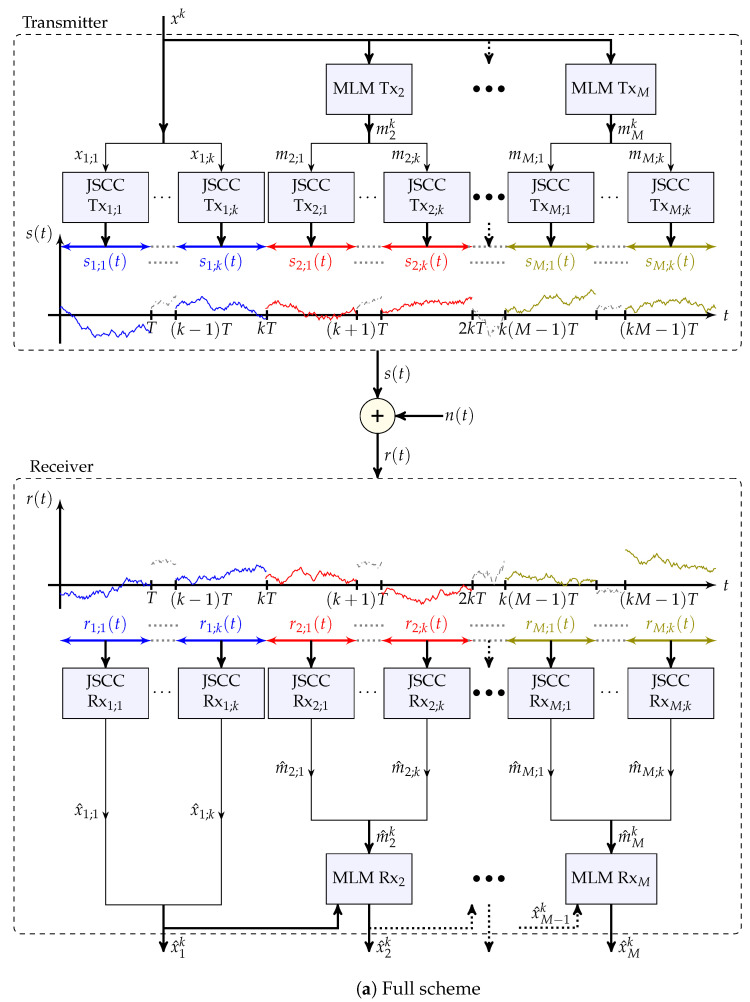
Block diagrams of Scheme 2 and of this scheme with the effective additive noise channels of Remark 6.

**Figure 5 entropy-25-01522-f005:**
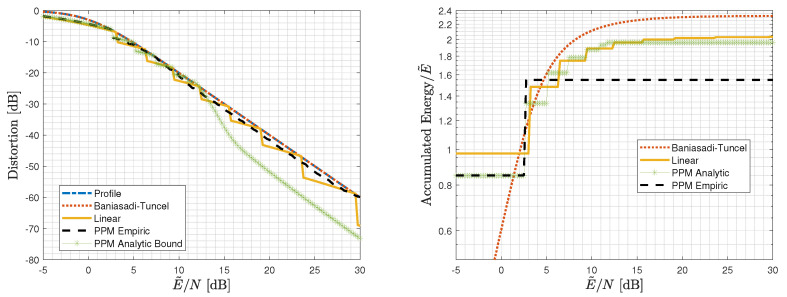
Distortion and accumulated energy of the layers utilized by the receiver at a given $\tilde{E}/N$ for a Gaussian source in the infinite-blocklength regime for a quadratic profile: Scheme 2 with linear layers with energy allocation $E_{i}=\Delta e^{-\alpha i}$ for Δ=0.975 and α=0.65. Empirical performance of the scheme with a linear layer with energy $E_{1}=0.85$ and an analog PPM layer with energy $E_{2}=0.75$, and analytic performance of the scheme of Theorem 4 with the parameters from its proof, and analytic performance of the scheme of Baniasadi and Tuncel [18].

**Figure 6 entropy-25-01522-f006:**
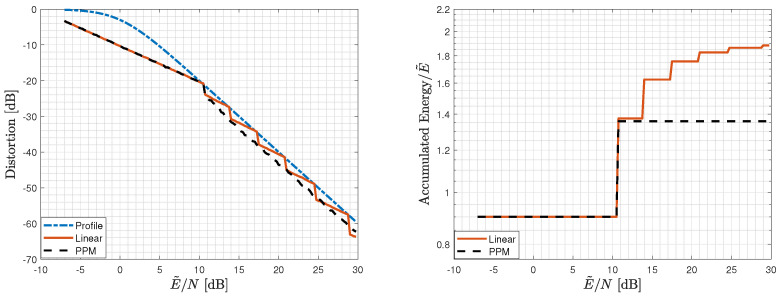
Distortion and accumulated energy of the layers utilized by the receiver at a given $\tilde{E}/N$ for a uniform scalar source for a quadratic profile: Scheme 2 with linear layers with energy allocation $E_{i}/\tilde{E}=\Delta e^{-\alpha i}$ for Δ=0.9 and α=0.64, and with a linear layer with energy $E_{1}=0.9\tilde{E}$ and an analog PPM layer with energy $E_{2}=0.346\tilde{E}$. The η value was optimized according to (Equation 76).

## Data Availability

Not applicable.

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
