# Peer review of "Energy-Limited Joint Source–Channel Coding of Gaussian Sources over Gaussian Channels with Unknown Noise Level"

_entropy, 2023, doi:10.3390/e25111522_

Round 1
Reviewer 1 Report
Comments and Suggestions for Authors
This paper studied the problem of JSCC over an energy-limited channel with unlimited bandwidth and/or transmission time when the noise level is unknown at the transmitter. They proposed MLM-based schemes that outperform the existing schemes thanks to the improvement in the performance of all layers with the ENR. Furthermore, by replacing (some of the) linear layers with analog PPM ones, further improvement was achieved. They further demonstrated numerically that the MLM-layered scheme works well in the scalar-source regime.
Strong aspects:
1. This paper proposes an effective solution for low-latency communication scenarios under limited energy constraints, with significant practical applications in the future.
2. This paper extensively derives and proves fundamental theories, providing ample theoretical groundwork for the proposed MLM-based layered scheme and PPM module.
Week aspects:
1. I believe that for individuals not well-versed in this field of research, this article is challenging to follow. The initial portion of the article is filled with numerous equations, lacks clarity in its structure, and contains many unfamiliar inferences. These factors gradually diminish the overall understanding of the article's main points during the reading process.
2. I suggest that the author include clear diagrams in the early part of the article to enhance the illustration of the main framework of the study. Additionally, optimizing the content of the theoretical derivation section to help readers focus on the most critical aspects would be beneficial.
Reviewer 2 Report
Comments and Suggestions for Authors
See the attached file

Reviewer 3 Report
Comments and Suggestions for Authors
In this paper, the authors suggest a universal scheme for transmitting an infinite sequence of Gaussian random samples over a continuous-time AWGN channel with infinite bandwidth, unknown variance, and a restriction on the input sample energy. The average squared distortion measure is used as a criterion.
Remarks.
1. In the introduction, the terms ENR, SNR, E, and energy polynomial profile (3a) are used without definitions. Their relation should be clarified.
2. How SDR (8) is related to SNR and ENR?
3. Eq. (6) determines continues-time cannel however eq. (15) determines discrete-time channel. What channel is studied in this paper?
4. In Section 3. Background: Modulo-Lattice Modulation, the potential reader knows that the authors are going to consider an application to Wyner-Ziv coding. In my opinion, the relation with this scheme should be explained earlier in the text.
5. It is not clear what results in Section 3. Background: Modulo-Lattice Modulation are known and what results are new ones. I did not find any discussion related to the application to Wyner-Ziv coding (this was claimed at the beginning of the section).
6. The authors’ contribution in Section 4. Background: Analog Modulations in the Known-ENR Regime is also not clear.
7. The multi-layer lattice is not described precisely, only the reference to the proof of its existence is given. It means that the claim ``we construct a universal scheme..’’ in the abstract is not correct. There is no any construction in the paper, only its existence is declared.
Typos:
1. Row 58: ``The differential entropy of a continuous random with probability density function f is defined…’’ should be ``random variable’’ (word ``variable’’ is missing)
In my opinion, both authors’ contribution and studied source-channel model should be clearly explained. I cannot judge the originality and the scientific value of the obtained results until a major revision has been made.
Reviewer 4 Report
Comments and Suggestions for Authors
Please find my comments attached as PDF document.

Please find my comments attached as PDF document.
Round 2
Reviewer 2 Report
Comments and Suggestions for Authors
The paper is significantly improved compared to the original version.
Most of my remarks are properly addressed.
However, I still do not understand interpretation of bound (7)
and the main result (74) if $\tilda E$ is defined as an arbitrary design parameter.
The authors refer to [7] but the notations and the form of expressions in [7] are not the same.
Moreover, in [7] bounds for different profiles are expressed in terms of (unknown) noise level.
I think, before the paper will be published, the authors should
give an understandable interpretation of the previous and new results.
Reviewer 3 Report
Comments and Suggestions for Authors
The authors significantly improved the introduction. The main contribution as well as results are still not clear to me.
1. In the introduction to Section 5 “Main Results”, the authors summarize the main contributions of the paper. It is explained that the new JSCC scheme is proposed. In this scheme, the successive refinement coding is replaced with modulo-lattice modulation (MLM). Further, the last layer is replaced with an analog pulse-position modulation (PPM). Then the authors write that “analytic and empirical results of the proposed scheme” will be compared. However, it is not explained what analytic and empirical results of the proposed scheme are analyzed. Is the main result a JSCC scheme or method of its analysis? What is the relation of the obtained performance to the same performance of previously used schemes?
2. What scheme was simulated? What are the system parameters? A detailed description is required.
3. Theorems 5.1 and 5.2 are achievability theorems which do not answer the questions in points 1 and 2.
Typos:
- p.1 ``sampleand’’ should be ``sample and’’
- Definition 3.2: ``quantizere’’ should be ``quantizer’’
- ``voronoi cell’’ should be ``Voronoi cell’’
- p. 10 Remark 1: ``value asymptotic value’’ should be ``asymptotic value’’
In my opinion, the main problem of the paper is that the suggested scheme is not reproducible because of the non-understandable description based on numerous citations. I cannot recommend the publication of this paper in its present form.
Round 3
Reviewer 3 Report
Comments and Suggestions for Authors
In my opinion, this paper can be published in its present form